# Site-Specific Microbial Decomposer Communities Do Not Imply Faster Decomposition: Results from a Litter Transplantation Experiment

**DOI:** 10.3390/microorganisms7090349

**Published:** 2019-09-12

**Authors:** Alessia Bani, Luigimaria Borruso, Kirsty J. Matthews Nicholass, Tommaso Bardelli, Andrea Polo, Silvia Pioli, María Gómez-Brandón, Heribert Insam, Alex J. Dumbrell, Lorenzo Brusetti

**Affiliations:** 1Faculty of Science and Technology, Free University of Bozen/Bolzano, Piazza Università 5, 39100 Bolzano, Italy; luigimaria.borruso@unibz.it (L.B.); andrea.polo@unibz.it (A.P.); silvia.pioli@unibz.it (S.P.); 2School of Life Sciences, University of Essex, Wivenhoe Park, Colchester, Essex CO4 3SQ, UK; kjmatt@essex.ac.uk (K.J.M.N.); adumb@essex.ac.uk (A.J.D.); 3Department of Agrifood and Environmental Science, University of Florence, Piazzale delle Cascine 18, 50144 Florence, Italy; tommaso.bardelli@yahoo.it; 4Institute of Microbiology, University of Innsbruck, Technikerstraβe 25d, 6020 Innsbruck, Austria; Maria.Gomez-Brandon@uibk.ac.at (M.G.-B.); Heribert.Insam@uibk.ac.at (H.I.); 5Departamento de Ecoloxía e Bioloxía Animal, Universidade de Vigo, E-36310 Vigo, Spain

**Keywords:** microbial diversity, oak forest, litter decomposition, transplantation, home field advantage

## Abstract

Microbes drive leaf litter decomposition, and their communities are adapted to the local vegetation providing that litter. However, whether these local microbial communities confer a significant home-field advantage in litter decomposition remains unclear, with contrasting results being published. Here, we focus on a litter transplantation experiment from oak forests (home site) to two away sites without oak in South Tyrol (Italy). We aimed to produce an in-depth analysis of the fungal and bacterial decomposer communities using Illumina sequencing and qPCR, to understand whether local adaptation occurs and whether this was associated with litter mass loss dynamics. Temporal shifts in the decomposer community occurred, reflecting changes in litter chemistry over time. Fungal community composition was site dependent, while bacterial composition did not differ across sites. Total litter mass loss and rates of litter decomposition did not change across sites. Litter quality influenced the microbial community through the availability of different carbon sources. Additively, our results do not support the hypothesis that locally adapted microbial decomposers lead to a greater or faster mass loss. It is likely that high functional redundancy within decomposer communities regulated the decomposition, and thus greater future research attention should be given to trophic guilds rather than taxonomic composition.

## 1. Introduction

Forests cover 30% of the Earth’s terrestrial surface and are important for carbon (C) cycling [1]. C stored in plant biomass is eventually returned to soil through the decomposition of leaf or needle litter and deadwood [2]. This litter also represents a reservoir for vital nutrients such as nitrogen (N), phosphorus and potassium [3,4,5]. The upper layer of the litter is rich in soluble molecules while the deeper layers have an increased proportion of recalcitrant molecules such as lignin and tannins, at least until that litter is transformed to humus and soil [6]. Fungi are considered the primary litter decomposers due to their ability to fragment dead plant tissues, starting with the breakdown of lignin and cellulose [7]. Bacteria are secondary consumers as they use the smaller and simpler compounds that become available after fungal activity [7]. However, recent work has shown that bacterial genomes contain cellulase genes [8] and they are capable of cellulolytic activity [9]. Bacteria could probably dominate the first phase of decomposition, where more soluble compounds are still available [9,10]. Despite a long history of research focusing on the dynamics of litter degradation, the role microbial communities play is still debated, especially regarding their local adaptation to litter inputs.

Several studies have hypothesised that litter decomposes fastest in its natural environment (Home Field Advantage theory, HFA) as decomposer populations are locally adapted to environmental parameters such as temperature, moisture and soil chemical composition [11,12]. Several reciprocal transplantation studies (swapping litter between home and away sites) have been conducted to test this. While some studies did show a quicker mass loss in the home environment and a lower mass loss in the away sites (positive HFA) [11,13], others found no evidence of HFA [14,15,16]. When HFA was observed, the proposed explanation suggested that the microbial community was adapted to the dominant litter and, therefore, was specialised in the use of that substrate [11,17]. In contrast, when HFA was not observed, the proposed explanation was that the microbial decomposers could adapt quickly to the new litter inputs [15,16]. In a recent paper, Palozzi and Lindo (2018) [18] analysed the contrasting HFA results, and proposed possible explanations for this based on both theoretical and environmental reasons. One possible explanation is that since definitions of HFA vary, standardized measurements are not yet available and comparisons between studies using Ayres’ formula [11] and those analysing raw data [15] are not directly comparable [18]. Contrasting results could also be due to a difference in climate conditions, especially soil temperature and water availability, or whether soil had been transplanted along with the litter [18]. Additionally, HFA should be considered as a spectrum rather than a binary process, especially in forests where multiple types of nutrients were present [18].

Few studies have investigated the microbiota in litter transplantation experiments [15,19] as most focus on the mass loss and nutrient status of the litter [15,20]. Moreover, those studies that did examine microbial communities used basic fingerprinting techniques [15,16,19]. Next-generation sequencing approaches are now needed to provide a higher resolution coverage of individual species and/or genera across both common and rare taxa and to avoid simply describing the most abundant taxa present [21,22]. Illumina data could provide the necessary information to answer whether specialist or generalist taxa are the main effector of decomposition and whether the specialist taxa facilitate HFA. Here, maintaining the carbon source constant, oak litter (*Quercus petraea* (Matt.) Liebl), we studied the drivers of microbial decomposition allowing the environmental variables to change to answer one main question: are decomposition sites (proxy of the environmental variables) influencing the microbiota and, as a consequence, the litter mass loss dynamics? 

## 2. Material and Methods

The experiment was carried out in South Tyrol (Italy). Three different forest sites were chosen according to their dominant vegetation type in order to represent the three main forest types in South Tyrol. The first was an oak forest (*Quercus petraea* (Matt.) Liebl.) at 530 m above sea level (a.s.l.) in the Monticolo forest area. The other two sites were a beech forest (*Fagus sylvatica* L.) at 1000 m a.s.l. in San Genesio Atesino and a rhododendron bush (*Rhododendron ferrugineum* L.) at 1530 m a.s.l. in Renon. The Monticolo site will be referred to here as the “home site”, while Renon and San Genesio Atesino as “away sites”. Even if located in the same geographical area (Adige valley, nearby the city of Bolzano), sites differed not only in species dominance but also in soil pH, temperature and altitude (Table 1). The same sites were used in Bani et al. 2018 [16]. The away sites are located at a higher altitude and thus have a lower average annual temperature than the home site. Temperature was measured across the entire experiment in all sites using a HOBO onset datalogger (Onset computer corporation, Bourne, MA, USA). Sites were selected to maximise the possibility of finding very different microbial communities.

During autumn 2014, fresh fallen oak leaves were collected from the Monticolo site within a period of time with no rainfall. Leaves were stored in sterile plastic bags at 4 °C, leaving their natural humidity, until deposition (10 May 2015). In situ litterbags, manufactured from nylon textile (15.0 × 15.0 cm, mesh size 40 µm) (Manifattura Fontana, Macherio, Italy), were adopted for the transplantation experiment. Mesh size was chosen based on the work of Aneja et al. (2006) [19]. At each experimental site, 18 litterbags were randomly placed within the upper layer of the litter horizon (O_L_) containing 3 g of air-dried oak litter, left to be covered naturally by new deposited leaf litter. Three litterbags were collected at each of the six sampling dates over a period of 519 days: 22 May 2015 (10 days), 7 August 2015 (87 days), 17 November 2015 (189 days), 17 March 2016 (310 days), 7 July 2016 (422 days) and 12 October 2016 (519 days). At the point of sampling, each sample was placed in a sterile plastic bag to avoid cross-contamination and loss of material. Total weight was measured for each replicate and then each replicate was subsampled in three equal weighed parts to be used to measure the mass loss for chemical analysis and molecular analysis of the microbial communities, respectively. These subsamples were stored at −20 °C until the analysis. Fifty-four (3 study sites × 6 sampling dates × 3 samples) litterbags were analysed at the end of the experiment.

Chemical analyses were performed as reported by Bani et al., 2018 [16]. Resuming, samples were oven-dried at 105 °C and periodically weighed until a constant weight was reached. Dried mass was calculated based on the amount of humidity present in the subsample and then used to calculate the dry weight for the entire sample. Total C and total N were measured by an elemental analyser (Elemental Analyser Flash 2000, Thermo Scientific, Milan, Italy) using 1 mg of dried sample at 60 °C for 72 h [23] (Appendix A).

To extract DNA, litter subsamples were ground to fine powder with liquid nitrogen under sterile conditions. DNA extraction was performed with 0.25 g of sample using the PowerSoil DNA isolation kit (MoBio, Arcore, Italy) following the manufacturer’s instructions. The concentration of the eluted genomic DNA was determined on a Qubit fluorospectrophotometer in combination with a DNA HS assay kit (Invitrogen, Milan, Italy) and normalised to 1 ng µL^−1^ for all samples so they could be processed for the Illumina sequencing on an automated system.

A quantitative Real Time PCR (qPCR) was used to quantify 16S rRNA gene copy numbers for bacteria and ITS gene copy numbers for fungi. qPCR was performed with 1× IQ SYBR GREEN SUPERMIX (Biorad, Segrate, Italy) on a Rotor-Gene 6000 Real Time Thermal Cycler (Qiagen, Milan, Italy), used in combination with Rotor-Gene Series software (Qiagen, Milan, Italy). Each standard reaction, 20 µL of final volume, contained 1× IQ SYBR GREEN SUPERMIX, forward and reverse primers (0.8 µL of 200 nM each primer, see below), 0.4 mg mL^−1^ Bovine Serum Albumine (BSA) (Thermo Fisher Scientific, Milan, Italy), distilled water (RNase/DNase free, Gibco, UK) and 2 µL of 1:10 diluted DNA-extracts, and ten-fold diluted standard DNA. The standard DNA was constructed from a purified PCR product of known concentration from a pure culture of *Nitrosomonas europaea* and *Aspergillus terrestris* for bacteria and fungi, respectively. Primers were 1055f/1392r [24] for bacteria, and ITS3/ITS4 [25] for fungi. QuantiFluor dsDNA Dye was used to determine the stock concentration (gene copies µL^−1^) standard curve freshly prepared with ten-fold dilutions ranging from 10^2^ to 10^9^ copies µL^−1^. All standards and samples were run in duplicate, following the cycling conditions shown in Bardelli et al. (2017) [26] and De Beeck et al. (2014) [27] for bacteria and fungi, respectively. To check for product specificity and potential primer dimer formation, runs were completed with a melting analysis starting from 65 to 95 °C, with temperature increments of 0.25 °C and a transition rate of 5 s. The purity of the amplified products was also checked by the presence of a single band of the expected length on a 1% agarose gel stained with the DNA stain Midori Green (Nippon Genetics, Dueren, Germany) and visualized by UV-transillumination (Vilber Lourmat Deutschland GmbH, Eberhardzell, Germany).

To study the composition of the fungal and the bacterial communities in the oak litter, we used 2 × 300 bp MiSeq amplicon sequencing of the ITS2 region and 16S rRNA gene, respectively, following Illumina’s “16S Metagenomic Sequencing Library Preparation” protocol (https://support.illumina.com/documents/documentation/chemistry_documentation/16s/16s-metagenomic-library-prep-guide-15044223-b.pdf). For the ITS region, we selected the ITS3 and ITS4 primers [25] and for the 16S gene, the V3–V4 hypervariable region using the 341f and 805r primers [28] both modified with the required Illumina sequencing adaptors. PCR was conducted in a total reaction volume of 25 µl comprising 12.5 μL of REDTaq ReadyMix (Sigma-Aldrich Co, Gillingham, UK), 5 μL of each (1 µM) primer and 2.5 μL of DNA template and, when necessary, BSA was added to prevent inhibition. The thermal cycling condition for the 16S rRNA gene amplification involved initial denaturation at 95 °C for 3 min followed by 30 cycles each of 30 s at 95 °C, 30 s at 55 °C, 30 s at 72 °C and a final elongation at 72 °C for 5 min. The thermal cycling protocol for ITS amplification involved 95 °C for 3 min followed by 30 cycles each of 30 s at 94 °C, 30 s at 53 °C and 30 s at 72 °C and final elongation at 72 °C for 5 min. All the reactions were performed on an Applied Biosystems Veriti 96-well thermal cycler (Fisher Scientific, Loughborough, UK). After the final elongation, the PCR products were stored at 4 °C. The PCR products were purified using Agencourt AMPure XP PCR Purification beads (Beckman Coulter Ltd., High Wycombe, UK), following the manufacturer’s instructions. A second short cycle PCR was performed on 5 µl of the purified product to attach the Nextera XT indices. The reaction mix was 5 μL of Nextera i5 and i7 index, 25 μL of REDTaq ReadyMix, 5 µl of purified PCR product and 10 µl of PCR water (Bioline Reagents Ltd., London, UK). The thermal cycling conditions consisted of an initial denaturation step of 3 min at 95 °C followed by 8 cycles each of 30 s at 95 °C, 30 s at 55 °C and 30 s at 72 °C on an Applied Biosystems Veriti 96-well thermal cycler. After a final extension step of 5 min at 72 °C, PCR products were kept at 4 °C. PCR products were again purified using Agencourt AMPure XP PCR Purification beads. PCR products were quantified with PicoGreen dsDNA quantification assays (Thermo Fisher Scientific Inc., USA), on a POLAR star Omega plate reader (BMG LABTECH GmbH, Ortenberg, Germany). Nextera XT amplicons were then pooled in equimolar concentration and the concentration and length of the pooled amplicon was verified on an Agilent 2100 Bio-analyser (Agilent Technologies, Stockport, UK). Sequencing was conducted on an Illumina MiSeq at the University of Essex (Colchester, UK). 

Sequence data from this study are deposited in the SRA archive under the project accession number: PRJNA521096.

Amplicon sequences were analysed according to Dumbrell et al. (2017) [29]. Raw MiSeq output was converted to Fastq format using bcl2fastq. Sequences were trimmed (*q* < 20) with Sickle [30] and error corrected using SPAdes [31] implemented within BayesHammer [32]. Sequences were paired-end aligned using the PEAR algorithm [33]. VSEARCH [34] was used for chimera checks and picking operational taxonomic units (OTUs) based on 97% similarity. The resultant OTU sequences were assigned taxonomy using the Naïve Bayesian Classifier [35] via the QIIME pipeline [36] against the RDP database for bacteria and the UNITE database for fungi [37]. All singletons were removed before the analysis.

To test whether the richness (expressed as OTU richness) and qPCR data were influenced by decomposition site and time, ANOVA was applied followed by the Turkey test (HSD) if the condition of normality and equality of the variance were met. If the data were not normally distributed, the Kruskal–Wallis test followed by a Least Significant Difference test was applied. Linear models were applied to test in which site bacterial richness was higher and had a steeper increase with time. 

Since the main goal was to study the influence of the environment on the decomposer community, we focused on bacteria and only on the saprotrophic fungal community instead of the total fungal community assessing the trophic mode via FUNGuild [38] before proceeding with further analyses. Even if we considered our primary interest the saprotrophic community, all the following analysis had been applied also on the total fungal community and the results are presented as Appendix A. For the microbial communities, we applied a multivariate approach using generalised linear models (GLMs) to test which factors influenced the most abundant genera (relative abundance above 1%) for bacterial and fungal saprotroph communities. The tested variables were: time, site, litter temperature and chemical composition of the litter expressed as C:N or total N content. We used a forward selection starting with the simplest model, analysing single factors in each model, and then increasing the complexity of the model including more factors and the interaction between them. GLMs were chosen based on the lowest Akaike Information Criterion (AIC) score. AIC scores provide an estimator of the goodness of fit of the statistical model, while accounting for the number of parameters included; the lowest AIC score is used to select the best model [39]. The resulting models were compared with a likelihood-ratio test (ANOVA, Monte Carlo resampling, 999 bootstraps). To visualise the results, latent variable models (LVMs) were applied for both the communities using a negative binomial distribution of the data. Finally, to understand what type of interaction occurs between the bacterial and fungal (saprotrophic and total) communities, a Spearman rank correlation test was applied. 

All the analyses were conducted in R software. “*Car*” [40] and “*agricolae*” [41] packages were used for testing normality and homogeneity of the variance as well as to perform the ANOVA, HSD, Kruskal–Wallis and LSD tests. The taxonomic composition of communities was analysed with “*phyloseq*” [42]. For GLM analysis, we used the “*mvabund*” package [43] and for LVM, the “*boral*” package [44]. Finally, co-occurrences were computed with “*corrplot*” [45] and “*hmisc*” [46]. All graphs were created with the “*ggplot2*” library [47].

## 3. Results

### 3.1. Bacterial and Fungal Community Composition

Samples, with singletons removed, were rarefied to the lowest number in each dataset: 3712 for bacteria and 7025 for fungi. A total of 4294 unique bacterial OTUs were found, of which 244 OTUs were assigned to genus level, and with 104 OTUs accounting for >1% of all sequenced reads. For fungal microbiota, we were able to detect 1793 OTUs, of which 321 were assigned a genus. Of the 321 fungal genera recorded, 272 were assigned to a trophic guild. A total of 35 fungal genera could not be assigned to any trophic level, while 66 genera were pathotrophic, 26 symbiotrophic and 104 exclusively saprotrophic and 41 genera had a mixed trophic behaviour (Table 2). The saprotrophic group was the most abundant group in all the samples (Table 2, Appendix A). We focused our attention on the saprotrophs (also including any mixed guilds with saprotrophics) as they are the most important players in the decomposition process [7].

The bacterial community across sites was dominated throughout the experiment by the phylum of *Proteobacteria* (68% ± 7%), followed by *Actinobacteria* (13% ± 6%), *Bacteroidetes* (9% ± 3%) and *Acidobacteria* (6% ± 4%) (Figure 1A). Dominant classes were *Alphaproteobacteria* (46% ± 6%), *Actinobacteria* (15% ± 6%), *Betaproteobacteria* (11% ± 6%), *Gammaproteobacteria* (10% ± 5%), *Acidobateria_*Gp1(8% ± 4%) and *Sphingobacteria* (8% ± 3%) (Appendix A). At the genus level, the predominant taxa were: *Sphingomonas* (17% ± 5%), *Luteibacter* (5% ± 3%), *Methylobacterium* (5% ± 3%), *Massilia* (4% ± 2%), *Terriglobus* (4% ± 2%), *Hymenobacter* (3% ± 2%), *Mucilaginibacter* (3% ± 2%), *Pseudomonas* (3% ± 3%), *Rhizobium* (2.5% ± 1%), *Granulicella* (2% ± 1%), and *Burkholderia* (2% ± 1%) (Figure 1B). The genus *Sphingomonas* was constantly present in the bacterial community. Other genera such as *Methylobacterium* were mostly present at the earliest stages of decomposition. *Luteibacter* became more abundant in middle–late stages in the home site and in Renon, while it was absent in San Genesio Atesino. Conversely, *Terriglobus* was present in the home site and in San Genesio Atesino, but not in Renon.

In the fungal community, the phylum *Ascomycota* (71% ± 24%) dominated throughout, followed by a temporally increasing presence of *Basidiomycota* (28% ± 24%) especially in the Renon site (Figure 2A). At class level, the dominant taxa were: *Agaricomycetes* (29% ± 27%), *Leotiomycetes* (29% ± 21%), *Sordariomycetes* (19% ± 16%) and *Dothideomycetes* (14% ± 9%) (Appendix A). The dominant genera were: *Mycena* (15% ± 23%), *Sistotrema* (12% ± 24%), *Lachnum* (7% ± 10%), *Plectania* (6.5% ± 16%), *Cladosporium* (6% ± 8%) and *Lemonniera* (4% ± 9%) (Figure 2B). *Cladosporium* and *Lemonniera* were the dominant genera in the earliest phases at all the sites. *Sistotrema* and *Lachnum* were present only in the intermediate phase and almost exclusively in one of the away sites (*Sistotrema* in Renon and *Lachnum* in San Genesio Atesino). *Mycena* and *Plectania* dominated at the last time point across sites. Saprotrophic genera included *Sistotrema*, *Plectania, Lachnum, Lemonniera*, while *Mycena* and *Cladosporium* belonged to the pathotrophic groups (Appendix A; mixed behaviour included pathotrophic-saprotrophic and therefore these were included in the GLMs).

### 3.2. Bacterial and Fungal Quantification

The microbial communities were quantified by qPCR of the 16S rRNA genes (Appendix A) and ITS region (Appendix A), respectively. The initial bacterial communities were quantitatively similar across the three sites with an abundance in the range of 10^9^ 16S rRNA gene copy numbers per gram of dry litter, while fungal ITS gene copy numbers were usually an order of magnitude higher. Only the fungal ITS gene copy numbers at the Renon site were significantly influenced by time (ANOVA d.f. = 5, F = 7.22, *p* < 0.05), while for all the other qPCR results, neither time nor decomposition site were statistically significant (Appendix A). The Fungi:Bacteria (F:B) ratio showed a similar trend over the three sites (Kruskal–Wallis: critical value = 14.52, *p* < 0.05 for Monticolo, critical value = 13.26, *p* < 0.05 for Renon and critical value = 11.43, *p* < 0.05 for San Genesio Atesino). At the early time points, fungi dominated the microbiome, while in the later sampling point, the F:B ratio was approximately one as a result of an increase in bacterial abundance and a decrease in fungi, especially after 310 and 422 days (Figure 3).

### 3.3. Mass Loss and Litter Chemical Composition Dynamics

The mass loss was approximately 60% of the initial mass across sites. No differences were found in the decomposition rates or in the remaining mass loss across the experiment (see Appendix A, ANOVA site *p* = n.s.). Similarly, the C:N ratio decreased constantly over time until reaching the ratio of 40 (Appendix A, ANOVA time *p* < 0.01, d.f. = 5, F = 25.57). However, in the San Genesio Atesino site, after 310 days, the C:N ratio increased as a consequence of a decrease in the amount of N present in the litter. Generally, in the entire experiment the total content of N (Appendix A) increased through time almost doubling the total N present at the end of the experiment (ANOVA time *p* < 0.01, d.f. = 5, F = 30.21). Total C decreased in samples from San Genesio Atesino and Renon to almost 50% of the initial amount but not in Monticolo litter, where it was almost constant across the entire experiment (Appendix A, ANOVA site *p* = 0.012, d.f. = 2, F = 6.02).

### 3.4. Bacterial and Fungal Community Structure

Bacterial (OTU) richness increased over time in all sites (Figure 4A), reaching maximum levels at the end of the experiment (HSD groups Monticolo: d.f. = 5, F = 18.640, *p* < 0.01; San Genesio Atesino: d.f. = 5, F = 9.517, *p* < 0.01; Renon: d.f. = 5, F = 2.145 *p =* 0.129 not significant). Home site samples showed the steepest increase of richness over time, as based on the linear model and the best model fit, based on R^2^. Renon data had the lowest R^2^ (Monticolo: slope = 287.80, R^2^ = 0.79, F = 59.360; Renon: slope = 191.2, R^2^ = 0.44, F = 12.520; San Genesio Atesino: slope = 232.57, R^2^ = 0.76, F = 51.550) while, in the away sites, richness increased more gradually until a stabilisation at the experiment end (422 and 519 days respectively). For saprotrophic fungi, OTU richness remained more stable over time (Figure 4B), with a general decrease in the second year of the experiment. The overall fungal richness was similar to the saprotrophic taxa (Appendix A). Interestingly, for both bacterial and saprotrophic taxon richness, time was statistically the most significant factor (bacteria: d.f. = 5, F = 18.64, *p* < 0.05 for Monticolo and d.f. = 5, F = 9.517, *p* < 0.05 for San Genesio Atesino; saprotrophic: d.f. = 5, F = 4.52, *p* < 0.05 for Monticolo and d.f. = 5, F = 5.20, *p* < 0.05 for San Genesio Atesino) except for the Renon site (bacteria: d.f. = 5, F = 2.145, *p* > 0.05 and saprotrophic: d.f. = 5, F = 0.30 *p* > 0.05).

To test the hypothesis that time and site influenced the microbiota, a multivariate approach was applied using General Linear Models (GLMs). The best GLM for bacteria was the more complex, including site, time, temperature and chemical composition of the litter expressed as C/N (Table 3, model 12). For saprotrophic fungi, even if AIC scores were similar, the best model included only time and site, but not their interaction (Table 3, model 7, Appendix A for total fungal community). For both groups, the strongest single predictor was always time (Table 3, model 1). Time influenced 74 out of 87 bacterial genera and 31 out of 134 saprotrophic fungal genera (bacteria: d.f. = 5, Dev = 3675, *p* < 0.05; saprotrophic: d.f. = 5, Dev = 1365, *p* < 0.05). Site, instead, influenced 12 bacterial and 10 saprotrophic genera (bacteria: d.f. = 2, Dev = 469.3, *p* < 0.05; saprotrophic: d.f. = 2, Dev = 375, *p* < 0.05, Appendix A for bacteria and Appendix A for saprotrophic fungi). Total N significantly influenced five bacterial genera (*Actinoplanes*, *Asticcacaulis*, *Dyadobacter*, *Gemmata* and *Niastella*), while three fungal genera were influenced by C:N (*Lemonniera*, *Peniophora* and *Spirosphaera*). To visualise the results of the GLM models, bacterial and fungal community composition was analysed with a latent variable model (Figure 5). The latent variable ordination plot showed explicitly the mean–variance relationship and, therefore, overcame the problem of the overdispersion of the samples [48]. As with the GLMs, the LVM confirmed the influence of time and site on microbial composition. Finally, we tested how the microbial communities interact through a correlation analysis. The results were different when only the saprotrophic fungi were included or when all the most abundant fungal genera were used (Appendix A, only the significant *p* < 0.05 are shown). The saprotrophic group developed a more complex type of relationship, both positive and negative, compared to the total fungal community.

## 4. Discussion

### 4.1. Factors Explaining Microbial Composition

Our initial hypothesis that decomposition site and time were the main environmental variables that influence the microbiome was only partially met since the site effect described by the AIC scored relatively low for both bacteria and fungi (Table 3, model 2). In general, our best models still had high AIC scores, reflecting how challenging it is to disentangle the factors driving the microbial community. Even if litter chemistry is expressed as C:N or as total N content, it is widely recognised as an important factor to shape the microbiota and vice-versa [49,50,51,52]. Our data showed that saprotrophic fungal composition was not explained by these parameters. N-fixing bacteria such as *Sphingomonas* or *Luteibacter* could provide N that is known to be limited in the litter environment, while they could receive secondary metabolites from fungi [50]. This has recently been shown for deadwood colonisation [10] and in the rhizosphere where diazothrophs and mycorrhiza, together, promote the growth of the host plant [53]. Positive interactions between microbial species with different functional roles were reported to affect process rates in soil nutrient dynamics [54]. Similarly, in our system, these positive interactions could enhance leaf litter degradation and humus formation. However, a positive relationship between two species does not necessarily imply a direct interaction as it may only reflect the similar habitat requirements of the two taxa without affecting the process dynamics [55].

Environmental parameters (site as proxy for all of them) had a greater influence on fungal decomposers than bacteria. Priority effect, assembly history or order of arrivals could possibly explain the dominance of specific taxa [56] as *Lachum* and *Sistotrema* (Figure 2). In addition, the contingency of the fungal community has been ascribed to a limited dispersion of fungal spores in air across the same region [57]. Additionally, fungi are known to strongly compete through the production of secondary metabolites or through hyphal interaction [56] and, in this perspective, *Sistotrema* is classified as a strong competitor [58]. These characteristics could explain the abundance of *Sistotrema* in San Genesio Atesino, and its decline in abundance over time could be attributed to a chemical change in litter composition and the greater ability of other taxa, such as *Mycena,* to use molecules such as lignin [59]. Bacteria are usually less influenced by environmental variability as they include cosmopolitan and habitat generalists’ genera [60]. Usually, these bacteria show no preferences in habitat colonisation and are generally found in different environments, ranging from soil to water and stone artwork [61], where multiple niches are available [60]. For example, *Pseudomonas* has broad ability to degrade complex molecules, including polycyclic aromatic compounds [62], which are very common in forest litter.

### 4.2. Microbial Community and Mass Loss Relationship

Even though the microbiota was influenced by the site and specific saprotrophic genera showed site specificity, no evidence of either a faster decomposition rate or greater mass loss at the home site was found (Appendix A). Mass loss for oak litter was similar in all of the sites and no significant differences were found between the remaining mass across decomposition sites (Appendix A). This finding is in contrast with previous studies that observed a quicker mass loss of litter at the home site and attributed this to the presence of a specialised microbial community [11,12]. Compared to other studies, our experiment used a longer decomposition time. Moreover, these studies used fingerprinting analysis [11,13,17,19,20] and they could only imply the presence of a specialised decomposer community. Interestingly, Ayres et al. (2006) [63] focused on microbial community composition, testing whether the microbiota was specialised in the decomposition of a specific litter with the use of an inoculum. Despite some limitation of their experimental setup (duration of the experiment or the non-natural conditions), their results showed no support for a specialisation and no quicker mass loss in native litter with the original inoculum [63]. Recently, in a study on fungal composition through a complete transplantation experiment, researchers have found that fungi showed site-dependent composition and a strong HFA for two types of litters [64]. However, HFA is strongly context dependent and it has been reported that the bias is more pronounced in poor litter quality (oak as broadleaves is, instead, considered rich litter) since poor litter benefit from the richer outer environment. Additionally, in ecosystems with higher nutrient resources, such as forests, HFA should be considered as a continuum and not a binary process, while in a nutrient-limited ecosystem, it should be considered a way to overcome limitations [18].

In our study, the similar decomposition and mass loss rates found between different sites (home and away) indicated that the microbial community functionality was the key factor rather than the mere microbial composition. In previous studies, including two types of litter, it was shown that functionality (expressed as potential enzymatic activity) was decoupled from the microbial community pattern, and it was possible to highlight microbial functional redundancy [50,65]. The presence of the same function in more than one species could be the most likely explanation as to why different microbial composition had the same rate of decomposition at different sites. Different species of microbes commonly share the same functional role, especially in the saprotrophic community, where they are specialised in the decomposition of litter and are considered to almost completely colonise this niche [7]. Redundancy in fungal enzymatic activities is found across different biomes, while community composition is influenced at a smaller scale with an endemic population within the biomes [57].

## 5. Conclusions

The present study is one of the first to have explicitly studied the composition of decomposers in a transplantation experiment. Our study showed how fungal and bacterial decomposers behave differently, since bacteria are known to have a more cosmopolitan distribution [66], while fungi are less likely to be ubiquitous. However, even if a specialised community is present, this does not imply a quicker mass loss in the home environment. We conclude that decomposition models and HFA should focus more on functional groups than on taxonomic composition.

## Figures and Tables

**Figure 1 microorganisms-07-00349-f001:**
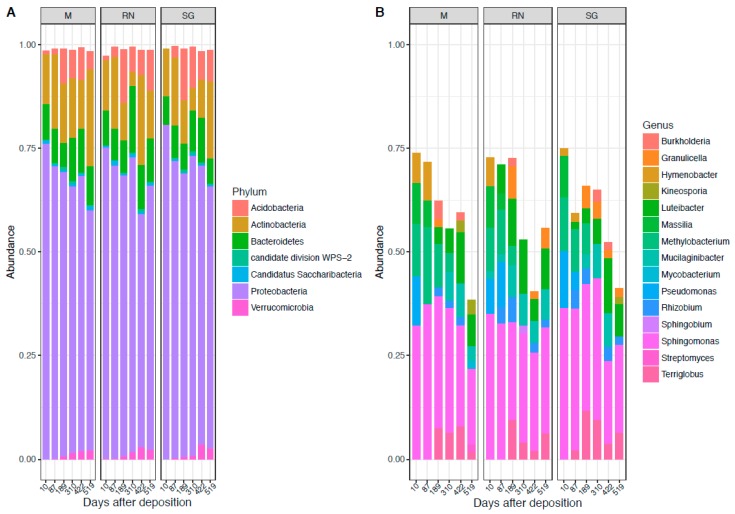
Bacterial composition at the phylum level (**A**) and genus level (**B**). Each panel represents a different site of decomposition: M, Monticolo; RN, Renon; SG, San Genesio Atesino.

**Figure 2 microorganisms-07-00349-f002:**
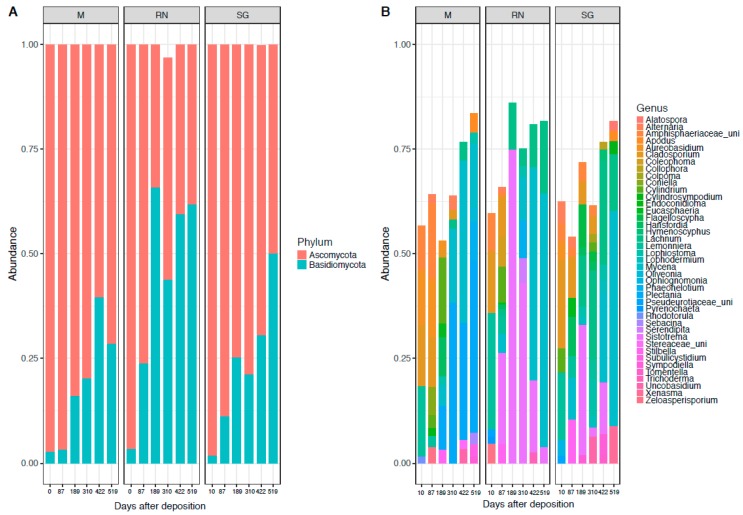
Fungal composition at the phylum level (**A**) and genus level (**B**). Each panel represents a different site of decomposition: M, Monticolo; RN, Renon; SG, San Genesio Atesino.

**Figure 3 microorganisms-07-00349-f003:**
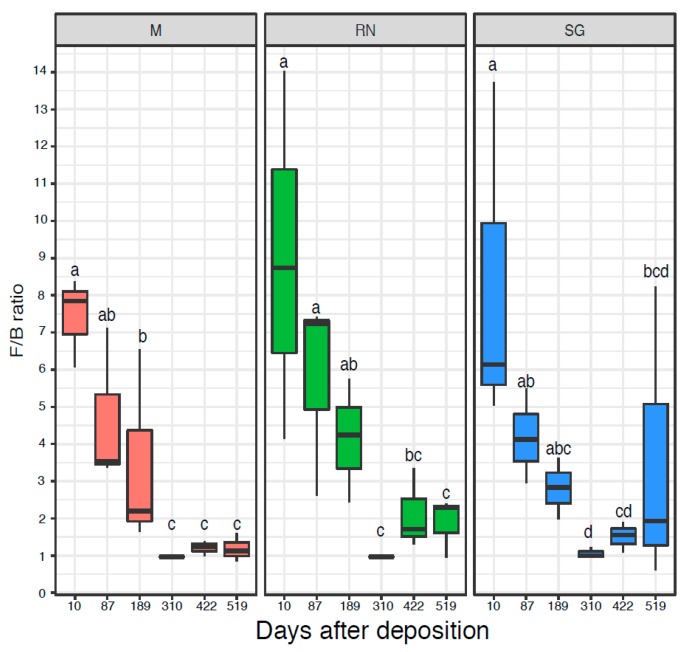
The Fungi:Bacteria ratio based on the number of copies of 16S and ITS. Each boxplot represents three samples. Each panel represents a different site of decomposition: M, Monticolo; RN, Renon; SG San Genesio Atesino. Different letters indicate significant difference as function of time effect (*p* < 0.05, Kruskal–Wallis followed by LSD test).

**Figure 4 microorganisms-07-00349-f004:**
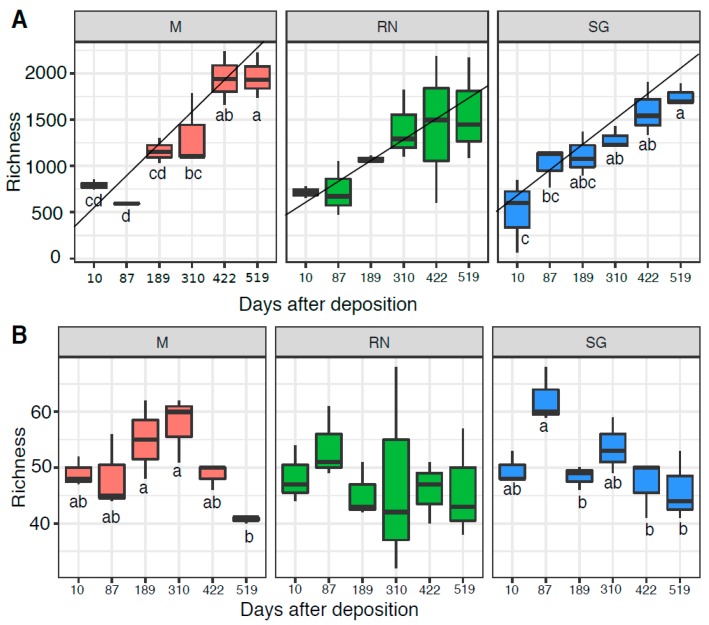
Richness of bacterial (**A**) and saprotrophic fungal (**B**) communities. Each boxplot represents three samples. Each panel represents a different site of decomposition: M, Monticolo; RN, Renon; SG, San Genesio Atesino. Different letters indicate significant difference as function of time effect (*p* < 0.05, ANOVA followed by HSD test). RN richness was not statistically different over time (*p* > 0.05).

**Figure 5 microorganisms-07-00349-f005:**
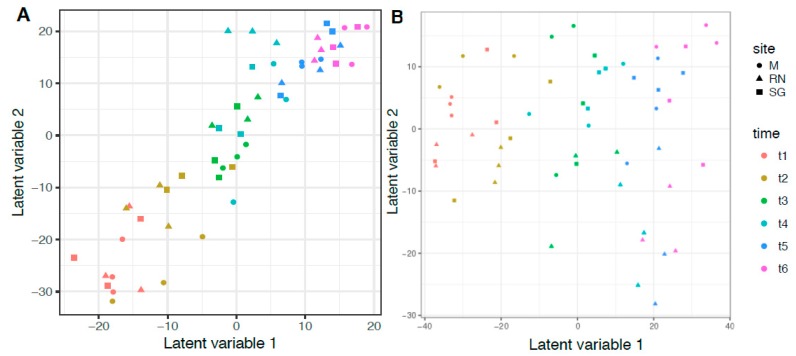
Latent Variable Model (LVM) ordination plot. (**A**) Bacteria genera composition and (**B**) Saprotrophic Fungi genera composition. Different colours indicate different time points, while different shapes indicate the different sites: M, Monticolo; RN, Renon; SG, San Genesio Atesino.

**Table 1 microorganisms-07-00349-t001:** Main characteristics of the selected study sites. a.s.l, above sea level.

Site Characteristic	Monticolo	San Genesio Atesino	Renon
Elevation	530 m a.s.l.	1000 m a.s.l.	1530 m a.s.l.
GPS coordinates	46°25′35″ N; 11°17′55″ E	46°32′35.4″ N; 11°18′36.4″ E	46°35′07.1″ N; 11°25′44.4″ E
Lithology	Quartz porphyritic	Quartz porphyritic	Quartz porphyritic
Soil type	Acid brown soil	Podzol	Podzol
Soil texture	Sandy loam	Sandy loam	Sandy loam
pH soil	5.50	6.13	4.83
Mean annual precipitation	800 mm	735 mm	970 mm
Mean annual temperature	11.8 °C (max. 21.3 °C, min. 3.6 °C)	10.5 °C (max. 17.0 °C, min. 3.0 °C)	6.8 °C (max. 14.3 °C, min. −0.7 °C)
Dominant vegetation	Oak (*Quercus petraea*)	European beech (*Fagus sylvatica*)	Rhododendron (alpine rose) (*Rhododendron ferrugineum*)

**Table 2 microorganisms-07-00349-t002:** Fungal trophic composition. The relative frequencies are calculated on the basis of all sample frequencies.

Trophic Mode	Number of Genera	Relative Frequencies
Pathotroph (Pat)	66	7.810245
Saprotroph (Sa)	104	53.52034
Symbiotroph (Sy)	26	9.91289
Pathotroph-Saprotroph (Pat-Sa)	10	0.8716193
Pathotroph-Symbiotroph (Pat-Sy)	8	1.627169
Saprotroph-Symbiotroph (Sa-Sy)	8	15.23502
Pathogen-Saprotroph-Symbiotroph (Pa-Sa-Sy)	1	0.03621281
Pathotroph-Saprotroph-Symbiotroph (Pat-Sa-Sy)	14	4.793488
Unassigned (U)	35	1.621936
Total	320	100

**Table 3 microorganisms-07-00349-t003:** Generalised linear models (GLMs) applied on the bacterial community and saprotrophic fungi community. For each model, the Akaike Information Criterion (AIC) score and whether the factor was statistically significant is provided. Models with * include the interaction between the factor, models including + instead do not account for interaction between the different factors. Best fitting models are highlighted in bold. *** *p* < 0.001; ** *p* < 0.005.

Model.	Variables	AIC Score Bacteria	AIC Score Saprotrophic
Null	1	46,568	28,094
Model 1	Time ***	43,920	27,094
Model 2	Site ***	46,530	27,963
Model 3	Total N ***	44,252	27,272
Model 4	Total C ***	46,031	33,242
Model 5	C/N ratio ***	44,126	27,272
Model 6	Temperature ***	45,989	33,065
Model 7	Time*** + Site ***	43,744	**26,908**
Model 8	Time *** × Site ** (Time × Site) ***	43,753	27,019
Model 9	Time *** × Site ** × Temperature ***	46,927	38,792
Model 10	Time *** × Site *** × C/N ***	**42,351**	35,163
Model 11	Time *** + Site *** + C/N *** + Temperature Time *** + Site *** + C/N *** + Temperature ***	43,160	30,818
Model 12	Time *** × Site *** × C/N *** + Temperature, T × S × C/N * Time *** × Site *** × C/N *** + Temperature ***, T × S × C/N ***	**42,558**	35,537
Model 13	Time *** + Site *** + TotalC ** + TotalN ** + C/N *** + Temperature *** Time *** + Site *** + TotalC *** + TotalN *** + C/N ** + Temperature ***	43,268	150,575

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
