# Peer review of "Site-Specific Microbial Decomposer Communities Do Not Imply Faster Decomposition: Results from a Litter Transplantation Experiment"

_microorganisms, 2019, doi:10.3390/microorganisms7090349_

Round 1

Reviewer 1 Report

Alessia Bani et al. presented the data from their long term studies analyzing microbial communities on decomposed oak litter at three elevation sites in South Italy. The results are part of a large study as it has been stated by authors in the Funding. In my opinion the text is interesting and valuable for publication, because refers  to the decomposition process and especially to the HFA  effect which need further scientific attention. Additionally, new modern approach as Illumina sequencing was used to better analyse fungal and bacterial communities.

Although I recommend the publication the manuscript need some attention and may be improved. 

Material and methods

- add geographical coordinates for each study site

- Table 1 describing study sites has been presented in the other author’s studies (Bani A. et al 2018. Microbial Decomposer Dynamics: Diversity and Functionality Investigated through a Transplantation Experiment in Boreal Forests), therefore authors can only refer to the published article or underline the similarities/differences between study sites in the text.

- were the litterbags only placed on the ground or were also covered by in situ litter?

- I would suggest to write “each sample” or “each litterbag’ instead of “each replicate” (line 104)

- Add “(3 study sites x 6 sampling dates x 3 samples) into the sentence “Fifty-four litterbags (3 study sites x 6 sampling dates x 3 samples) were analyzed at the end of the experiment.”

Results

Generally, results are clearly described but the graphs need to be improved.

- Figure S1, the full names on X axis such as “Pathogen-Saprotroph-Symbiotroph” are very small. Authors can use symbols “Pa-Sa-Sy”, “Pa”, “Pa-Sa” etc. on the graph and refer them in the table 2 as “Pathotroph (Pa)”

- Figure 2. It is difficult to identify the certain genus on the graph B, because of the colors used. I suggest to use other color palette or set the colors manually in ggplot to present better contrast.

- Figure S4 – present data with various line types ( solid, dashed, longdash) and geometric points (circle, triangle, squares) for three study sites.  The graph are only readable in color not in grayscale.

- Figure S6 and S7- the labels on the axes are unreadable and cannot be publish or presented in electronic version

Minor comments

-line 96: “whit” or “with”?

-line 101 – change “Three replicates litterbags”, into “Three litterbags” because  it is clear that on the same plot they are replicates

- figure S3b: “Dasy after deposition” change to “Days after deposition”

- axis labels on the same graph have different font size (Figure 5). Standardize the font size, graph lines color etc

-line 201 – please add “... in R software”

Author Response

Alessia Bani et al. presented the data from their long term studies analyzing microbial communities on decomposed oak litter at three elevation sites in South Italy. The results are part of a large study as it has been stated by authors in the Funding. In my opinion the text is interesting and valuable for publication, because refers  to the decomposition process and especially to the HFA  effect which need further scientific attention. Additionally, new modern approach as Illumina sequencing was used to better analyse fungal and bacterial communities.

Although I recommend the publication the manuscript need some attention and may be improved. 

Material and methods

- add geographical coordinates for each study site: added into table 1

- Table 1 describing study sites has been presented in the other author’s studies (Bani A. et al 2018. Microbial Decomposer Dynamics: Diversity and Functionality Investigated through a Transplantation Experiment in Boreal Forests), therefore authors can only refer to the published article or underline the similarities/differences between study sites in the text. Well, this is right, but the repetition of the table 1’s data has the meaning to avoid readers to download the other paper just to see the original table. On the other side, these are simply environmental data of very general interest.

- were the litterbags only placed on the ground or were also covered by in situ litter? Placed on ground and let new dead leaves to cover bags. This was added (line 104).

- I would suggest to write “each sample” or “each litterbag’ instead of “each replicate” (line 104): Done

- Add “(3 study sites x 6 sampling dates x 3 samples) into the sentence “Fifty-four litterbags (3 study sites x 6 sampling dates x 3 samples) were analyzed at the end of the experiment.”:  Done (line 112).

Results

Generally, results are clearly described but the graphs need to be improved.

- Figure S1, the full names on X axis such as “Pathogen-Saprotroph-Symbiotroph” are very small. Authors can use symbols “Pa-Sa-Sy”, “Pa”, “Pa-Sa” etc. on the graph and refer them in the table 2 as “Pathotroph (Pa)”. Done as suggested

- Figure 2. It is difficult to identify the certain genus on the graph B, because of the colors used. I suggest to use other color palette or set the colors manually in ggplot to present better contrast. Thanks for the suggestion. Done.

- Figure S4 – present data with various line types ( solid, dashed, longdash) and geometric points (circle, triangle, squares) for three study sites.  The graph are only readable in color not in grayscale. Done as suggested.

- Figure S6 and S7- the labels on the axes are unreadable and cannot be publish or presented in electronic version. With increased the resolution of writings.

Minor comments

-line 96: “whit” or “with”? With. Modified.

-line 101 – change “Three replicates litterbags”, into “Three litterbags” because  it is clear that on the same plot they are replicates. Done.

- figure S3b: “Dasy after deposition” change to “Days after deposition”. Done.

- axis labels on the same graph have different font size (Figure 5). Standardize the font size, graph lines color etc. Done.

-line 201 – please add “... in R software”. Done

Reviewer 2 Report

This is an interesting and well written paper.  I just had minor editorial comments that are included in an attached PDF.

Author Response

Line 66: space added

Line 78: GPS positions added into Table 1

Line 114: comma added

Line 114: Briefly has been replaced.

Line 115: Sentence has been changed.

Line 137: space added.

Line 231: “at the” added

Line 237: changed.

Lines 339-341: changes has been done accordingly.

Line 344-345: the two suggested changes has been done.

Line 354-355: the sentence has been corrected.

Line 364: include has been inserted because it is more suitable in this sense.

Line 367: the sentence has been changed.

Line 372: the two corrections has been done.

Lines 377-380: the changes has been added.

Line 385: the change has been done.

Lines 390-395: all the modifications has been included.

Line 397: done.

Line 560: done.